# Anti-Inflammatory Strategies Targeting Metaflammation in Type 2 Diabetes

**DOI:** 10.3390/molecules25092224

**Published:** 2020-05-09

**Authors:** Alina Kuryłowicz, Krzysztof Koźniewski

**Affiliations:** Department of Human Epigenetics, Mossakowski Medical Research Centre, Polish Academy of Sciences, 5 Pawinskiego Street, 02-106 Warsaw, Poland; krzychukoz@gmail.com

**Keywords:** metaflammation, insulin resistance, insulitis, type 2 diabetes, anti-inflammatory treatment

## Abstract

One of the concepts explaining the coincidence of obesity and type 2 diabetes (T2D) is the metaflammation theory. This chronic, low-grade inflammatory state originating from metabolic cells in response to excess nutrients, contributes to the development of T2D by increasing insulin resistance in peripheral tissues (mainly in the liver, muscles, and adipose tissue) and by targeting pancreatic islets and in this way impairing insulin secretion. Given the role of this not related to infection inflammation in the development of both: insulin resistance and insulitis, anti-inflammatory strategies could be helpful not only to control T2D symptoms but also to treat its causes. This review presents current concepts regarding the role of metaflammation in the development of T2D in obese individuals as well as data concerning possible application of different anti-inflammatory strategies (including lifestyle interventions, the extra-glycemic potential of classical antidiabetic compounds, nonsteroidal anti-inflammatory drugs, immunomodulatory therapies, and bariatric surgery) in the management of T2D.

## 1. Introduction

A sedentary lifestyle combined with increased consumption of more energy-dense food results in a situation when obesity has reached epidemic proportions globally. According to the World Health Organization reports, 1.9 billion adults are overweight, at least 650 million are clinically obese, and we observe an increasing number of obese individuals in all age categories both in industrialized societies and in developing countries [1]. Obesity is a major risk factor for type 2 diabetes (T2D), and trends in prevalence and incidence of both diseases are very similar. Epidemiological studies indicate that an obese individual has a sevenfold higher, while overweight—a threefold higher risk of T2D, when compared to normal-weight subjects; however, the relative risk depends on the study population characteristics [2]. In turn, approximately 80% of type 2 diabetic patients are overweight [3].

The maintenance of glucose homeostasis depends on normal insulin secretion by the pancreatic β cells and the normal sensitivity of tissues to insulin [4]. One of the concepts explaining the coincidence of obesity and T2D is the theory of chronic, low-grade inflammation, called metaflammation. It was based on the initial observation that obese individuals, compared to these with normal body weight, are characterized by the permanently elevated serum levels of inflammatory markers such as C-reactive protein (CRP), tumor necrosis factor α (TNFα) or interleukin 6 (IL-6), consequently replicated by other authors reviewed in [5].

This chronic inflammatory state might contribute to the development of T2D by increasing insulin resistance in peripheral tissues (mainly in the liver, muscles, and adipose tissue) and by targeting pancreatic islets and in this way impairing insulin secretion [6,7].

High prevalence of T2D and obesity constitute not only a medical but also a substantial economic and social problem [8], therefore, there is a constant need for novel therapeutic approaches targeting pathomechanisms common for both diseases. This review presents current concepts regarding the role of metaflammation in the development of T2D in obese individuals as well as data concerning possible application of different anti-inflammatory strategies (including lifestyle interventions, the extra-glycemic potential of classical antidiabetic compounds, nonsteroidal anti-inflammatory drugs, immunomodulatory therapies, and bariatric surgery) in the management of T2D.

It should be underlined, that the origin of obesity-related chronic inflammation is not entirely settled. Some animal studies suggest that high-fat-diet (HFD) changes gut microbiota and gut barrier function, leading to endotoxemia, lipopolysaccharide-induced activation of immune cells, and insulin resistance in rodents reviewed in [9]. However, several lines of evidence show that intrinsic changes in adipocytes function are crucial for the activation of inflammatory responses in adipose tissue. The infection-related metabolic changes will not be covered in this review.

## 2. Concept of Metaflammation

Patients with T2D manifest abnormalities in tissues’ insulin sensitivity (that predominantly refers to adipose tissue, muscle, and liver) as well as in pancreatic insulin secretion. Insulin is a chief inhibitor of lipolysis, and by inhibiting the hormone-sensitive lipase (LIPE) restrains the release of free fatty acids (FFA) from the adipocytes. The majority of T2D subjects are characterized by excessive adiposity, especially in the visceral region, where adipocytes have a high lipolytic rate and are primarily refractory to insulin [10]. Therefore, most of the T2D patients have chronically elevated plasma FFA levels that lead to insulin resistance in muscle and liver and impairs insulin secretion in the pancreas [11,12]. Another consequence of systemic high FFA concentrations in both obese and non-obese T2D individuals is increased storage of triglycerides in peripheral tissues, e.g., muscle and liver that correlates closely with their resistance to insulin [13,14]. This phenomenon has been referred to as “lipotoxicity” and also contributes significantly to β cell dysfunction [15].

Excess of FFA in the circulation together with the accumulation of lipids in tissues leads to the profound changes in cell metabolism, including, among others, mitochondrial dysfunction that contributes to the endoplasmic reticulum (ER) stress, hypoxia, cell hypertrophy, death and fibrosis reviewed in [16,17]. All of these processes result in the increased expression of genes encoding cytokines, chemokines, and adhesion molecules in the peripheral tissues what subsequently attracts infiltrating immune cells that additionally contribute to the synthesis of pro-inflammatory mediators impairing tissue function and in this way the vicious circle closes [7]. This low-grade, chronic inflammation originated from metabolic cells in response to excess nutrients is defined as metaflammation. Even though it can be considered as a defensive reaction, it occurs to be a double-edged weapon since it contributes to the multiorgan dysfunction and results in both: increased insulin resistance as well as in impaired insulin secretion. The concept of metaflammation is schematically presented in Figure 1.

Excess of nutrients in the course of obesity impairs adipocyte metabolism leading to mitochondrial dysfunction that contributes to the endoplasmic reticulum stress, hypoxia, and cell hypertrophy. These processes result in the increased expression of genes encoding cytokines, chemokines, and adhesion molecules in adipose tissue what subsequently attracts infiltrating immune cells (macrophages and different subsets of T cells) that additionally contribute to the synthesis of pro-inflammatory cytokines. Pro-inflammatory mediators (such as e.g., tumor necrosis factor-alpha, TNFα, and interleukins 1 and 6) impair adipose tissue function in the auto- and paracrine manner but also influence other tissues including liver, muscle, and pancreatic islets. In the liver and muscles, this process leads to the development of insulin resistance while in the pancreatic islands to disturbances in insulin secretion and apoptosis of β cells.

The excess of nutrients impairs adipocyte metabolism, alters adipokines secretion, and attracts inflammatory cells to adipose tissue. Even though the metabolic inflammation was first described in adipose tissue, it has to be emphasized that the obesity-related influx of immune cells occurs in many other tissues such as liver, muscle, and pancreatic islets. Moreover, the stromal components and metabolic cells, such as hepatocytes, myocytes, and β cells produce mediators (e.g., cytokines and chemotactic molecules) that influence immune cells’ action. These bidirectional interactions between immune, metabolic, and stromal components of particular organs are critical in determining physiological and pathological outcomes [7]. In the liver and muscles, this process leads to the development of insulin resistance while in the pancreatic islands to disturbances in insulin secretion and apoptosis of β cells.

### 2.1. Inflammation of Adipose Tissue

Given the close relationship between visceral adiposity and metaflammation, obesity-related changes in adipose tissue metabolism seem to play a pivotal role in this phenomenon. Studies performed in cell lines, animal models of obesity, and human tissues suggest that adipocytes hypertrophy is a primary trigger for the activation of the pro-inflammatory pathways in adipose tissue [16,18]. Recent evidence suggests that mitochondria are essential for maintaining metabolic homeostasis in adipocytes, and their dysfunction in the course of obesity (reflected by downregulation of cellular pathways involved in fatty acid oxidation, ketone body production and breakdown, and the tricarboxylic acid cycle) is directly associated with the intensity of inflammation and insulin resistance [19,20]. Mitochondrial dysfunction resulting in decreased fatty acid oxidation increases triglyceride accumulation. Moreover, it may trigger cell death in adipocytes and contribute to a defective differentiation of preadipocytes to mature adipocytes and adipose tissue fibrosis reviewed [17,21].

An important intracellular signaling cascade involved in the development of inflammatory response in the adipose tissue is the nuclear factor κB (NF-κB) pathway. In mammals, the NF-κB family of transcription factors consists of five proteins: NF-κB1, NF-κB2, RelA, RelB, and c-Rel that form homo- and heterodimers, that differ in their ability to regulate gene transcription. In unstimulated cells, NF-κB dimmers are sequestered in the cytoplasm thanks to their association with κB inhibitors (IκBs). Excess of FFA (as well as other stimuli) triggers the transduction pathways leading to the dissociation of NF-κB from IκB proteins. That involves IκB phosphorylation and subsequent dissociation. Dissociation of IκB exposes the nuclear localization signal in NF-κB proteins and leads to their nuclear translocation and binding to the promoters of target genes, including those involved in the inflammatory response: cytokines (e.g., IL-1, IL-6, IL-12, TNFα), chemokines (e.g., IL-8), adhesion molecules (e.g., selectin E, ICAM1, VCAM-1) and many others that attract immune cells [22]. An essential role in both: NF-κB activation and chemoattraction play leukotrienes (LTs) produced by hypertrophic adipocytes [23].

A predominant immune cell type that accumulates in adipose tissue during the course of obesity are macrophages that constitute 30–50% of the non-adipocyte cell fraction [24]. Both animal models of diet-induced obesity and studies performed in human tissues suggest that the intensity of the macrophages infiltration is higher in the visceral adipose tissue (VAT) compared to the subcutaneous (SAT) depot, however not all studies are unanimous in this respect [25,26,27]. Experiments with animals on a high-fat diet (HFD) suggest that the majority of these macrophages are recruited from the circulation; however, expansion of the resident macrophage population also contributes to the formation of the inflammatory infiltration in the adipose tissue of obese individuals reviewed in [14]. In humans, the number of macrophages infiltrating adipose tissue increases with its amount; however, until now, their origin remains unknown [24]. Macrophages isolated from the adipose tissue of obese individuals also differ in their secretory profile, which can be defined as pro-inflammatory, compared to macrophages present in adipose tissue of normal-weight subjects [28]. What is interesting, both animal and human studies suggest that the intensity of macrophages infiltration and their pro-inflammatory properties in adipose tissue not always decrease with weight loss and improvement of metabolic parameters [29,30].

Apart from macrophages, other types of immune cells infiltrate adipose tissue in the course of obesity and contribute to the development of metaflammation [31]. Among them are different subpopulations of lymphocytes, and their relative proportions change with increasing adiposity [32].

Infiltration of adipose tissue by T cells precedes pro-inflammatory macrophages and seems to be necessary for their subsequent recruitment and activation. In mice, HFD leads to a shift in the T cells populations towards Th1 and cytotoxic T cells with a relative loss of regulatory T cells (Tregs) with the onset of obesity, which may be a causative mechanism of subacute, chronic inflammation maintenance. What is important, the presence of Tregs in adipose tissue is positively associated with insulin sensitivity and improvement of glucose homeostasis in obese mice [33].

In humans, a systemic imbalance in Th1/Th2 response has been linked with disturbances in glucose homeostasis leading to the development of insulin resistance and T2D [34]. T2D patients are characterized by the elevated concentrations of cytokines belonging to Th1 responses (such as e.g., IFN-γ and TNFα) with a relative suppression of Th2 and Tregs-related immunosuppressive cytokines (such as e.g., IL-4, Il-10, and IL-10, IL-35, respectively). The enhanced Th1 profile with suppression of Th2 and Tregs cytokines correlates with the intensity of oxidative stress, insulin resistance, and development of micro- and macrovascular complications such as retinopathy, nephropathy and coronary artery disease reviewed in [35]. However, studies regarding changes in subpopulations of T cells infiltrating adipose tissue in human obesity are not univocal, and the role of local Tregs deficiency in the development of metaflammation remains to be clarified [36,37].

Immune cells infiltrating adipose tissue during the course of obesity are a primary source of the pro- and anti-inflammatory cytokines and chemokines, however, adipocytes themselves and other stromal cells also contribute to this phenomenon [38]. Increased expression of genes encoding, e.g., TNFα, IL-1β, IL-6, IL-18 and monocyte chemoattractant protein-1 (MCP-1) is consistently observed in adipose tissue of mice on HFD and mice genetically predisposed to obesity (e.g., in Lep_ob/ob_ a Lepr_db/db_ mice with functional mutations of genes encoding leptin and leptin receptor, respectively) [39]. Similar phenomena take place in human adipose tissue and are associated with a moderate elevation of inflammatory markers in sera reviewed in [16]. However, it has yet to be determined, which adipose tissue depot: SAT or VAT, plays a dominant role in these processes. As was mentioned before, morphological studies comparing the intensity of inflammatory infiltration in different adipose tissue depots suggest that in both obese and normal-weight individuals, VAT samples contain more macrophages than SAT [25,26,27,39,40]. However, studies that correlated serum cytokine concentrations with the incidence of obesity-related diseases and fat distribution [41,42] or with mRNA levels for various cytokines [43] did not give an unambiguous answer to this question. Based on our results and findings of other research groups, one can conclude that SAT is a more potent place of cytokine synthesis than VAT since concentrations of, e.g., pro-inflammatory IL-1β, IL-6, and IL-15 are significantly higher in SAT than in VAT independently from the weight. However, obesity is associated with the increased pro-inflammatory activity of VAT, especially in individuals meeting the diagnostic criteria for the metabolic syndrome [44]. However, there are reports that question the dominant role of VAT in the development of metaflammation, including a study showing that the mRNA levels of several pro-inflammatory interleukins were higher in SAT than in VAT of obese patients [45]. Indeed, central adiposity is commonly associated with high amounts of abdominal SAT, which has a unique gene expression profile, different from the SAT collected from other depots, e.g., in the gluteofemoral area [46].

Pro-inflammatory cytokines synthesized in adipose tissue influence whole body function, but in the auto- and paracrine manner, have an impact on adipocytes themselves. For instance, both TNFα and IL-6 induce insulin resistance in rodents and block insulin action in murine (3T3-L1) adipocytes. However, Il-6 can also induce oxidation of FFA and adipose tissue browning in animal models of obesity [47]. IL-1β seems to play a central role in macrophage-adipocyte crosstalk, which impairs insulin sensitivity in adipose tissue by inhibition of insulin signal transduction. Moreover, IL-1β stimulates the production of IL-8, a potent chemoattractant involved in the adhesion of monocytes to endothelium and, in the migration of vascular smooth cells, proposed, therefore, as a mediator between obesity and atherosclerosis reviewed in [48].

Mitochondrial dysfunction associated with excessive adiposity is also associated with dysregulation of adipokines secretion. This concerns, e.g., adiponectin, a protein hormone almost exclusively produced in adipocytes that has anti-inflammatory, antiatherogenic, and anti-oxidative properties reviewed in [17]. Adiponectin also exhibits insulin-sensitizing effects and may increase insulin secretion [49]. In several studies, adiponectin levels measured in serum and adipose tissue of obese individuals were significantly lower compared to the normal-weight subjects [50]. On the contrary, obesity is accompanied by an increased secretion of leptin. The serum level of this adipokine serves as a gauge for energy reserves and directs the hypothalamus to adjust food intake and energy expenditure. Both obese animals and humans are characterized by resistance to leptin since its high serum levels do not translate to anorectic effects. The role of leptin and leptin resistance in the development of T2D is composed since, on the one hand, leptin can decrease insulin secretion acting directly on the pancreatic β cells and increase lipid accumulation in the liver, while on the other enhance glucose uptake and oxidation in skeletal muscle. The important mechanism linking obesity-related hyperleptinemia and insulin resistance is related to pro-inflammatory properties of leptin that involve modulation of T cells action and upregulation of multiple inflammatory cytokines (including TNFα, IL-6) reviewed in [51].

The inflammatory process in the adipose tissue impairs adipocyte metabolism and is regarded as a chief mechanism linking increased adiposity with insulin resistance. This phenomenon has been confirmed by several animal models of diet-induced obesity. For instance, in C57BL/6 mice, even three days of HFD (60% kcal from fat) lead to an increase in adipocyte number and size parallel to the glucose intolerance, hyperinsulinemia, and systemic insulin resistance [52]. However, in animal models, the development of insulin resistance in response to HFD is highly heterogeneous, and a total fat mass is not its direct predictor, suggesting that changes that take place in other tissues may play a role in determining insulin sensitivity [53].

### 2.2. Inflammation in the Liver

It is estimated that approximately 70%–80% of obese and diabetic patients have non-alcoholic fatty liver disease (NAFLD) that is associated with more severe insulin resistance, hyperinsulinemia, and dyslipidemia [54]. In the diabetic patient in the basal state, the liver represents a major site of insulin resistance, and this is reflected by the overproduction of glucose despite the presence of both fasting hyperinsulinemia and hyperglycemia. Lipid accumulation in the liver results from the imbalance between the delivery of lipids and their hepatic uptake, synthesis, oxidation, and secretion, and several studies suggest that mitochondrial dysfunction plays a key role in the development of advanced NAFLD and its progression to non-alcoholic steatohepatitis (NASH) [55]. The pathogenesis of the NAFLD-associated insulin resistance is composed. On the one hand, cytokines secreted by the adipose tissue (namely TNFα, IL-1β, and IL-6) down-regulate liver insulin sensitivity by the activation of pro-inflammatory pathways parallel to the inhibition of insulin receptor signaling in the hepatocyte. On the other hand, excessive hepatic lipid accumulation followed by hepatocyte dysfunction promotes infiltration of the liver by the immune cells, mainly macrophages, but also natural killer cells and T-cells. Infiltrating macrophages recruit both from the resident liver population (Kupffer cells) and the bone marrow, and, as in the case of adipose tissue, are the primary source of the pro-inflammatory cytokines that contribute to the development of insulin resistance and progression of NAFLD to the non-alcoholic steatohepatitis (NASH) and subsequently to cirrhosis and hepatic carcinoma. This polarization of macrophages population from the non-inflammatory, secreting IL-4 and IL-13 (also known as “alternatively activated” or M2) towards pro-inflammatory, secreting IL-1β and TNFα (“classically activated” or M1) is a hallmark of metaflammation [56]. The critical role of TNFα in the development of NAFLD-related insulin resistance has been proved in animal and human studies, and its neutralization can substantially reduce hepatic steatosis in genetically obese (*ob*/*ob*) mice. HFD also leads to the increased expression of IL-1α/β in the liver, and knockout of these two cytokines protects from inflammation related to diet-induced steatosis in experimental animals. Therefore IL-1 inhibitors are being considered as a therapeutic option for NAFLD treatment in humans reviewed in [57]. Besides, obesity-associated hypoadiponectinemia and hyperleptinemia also contribute to these phenomena by impairing FFA metabolism and promoting chronic liver inflammation [58,59].

### 2.3. Inflammation in the Muscles

Skeletal muscles (SM) are the primary site of glucose disposal in man, responsible for approximately 80% of total body glucose uptake [4]. Studies with the euglycemic insulin clamp proved that even in lean T2D patients, response to a physiologic increase in plasma insulin concentration is delayed, and the ability of the hormone to stimulate glucose uptake is blunted compared to non-diabetic controls [60]. Severe muscle resistance to insulin in diabetic subjects was confirmed by the positron emission tomography (PET) scanning [61]. The pathogenesis of insulin resistance in skeletal muscle is complex and involves intramyocellular lipids deposition, mitochondrial defects, including reduced oxidative capacity and metabolic inflexibility, as well as the influence of the adipokines secreted by the malfunctioning adipose tissue in the course of obesity [62,63]. However, as in the case of the liver, in the overnutrition state, inflammation occurs also in the skeletal muscles themselves, and in auto- and paracrine manner contributes to their insulin resistance.

Like adipocytes, myocytes secrete several cytokines, known as myokines, including not only IL-6, IL-8, and TNFα but also irisin, myonectin, and myostatin. Expression of genes encoding most of the myokines is regulated by exercise and muscle contraction, and their effects on glucose and lipid metabolism, as well as on inflammation can be beneficial reviewed in [64]. For instance, IL-6 increases glucose uptake, lipolysis, and oxidation of FFA and mediates anti-inflammatory effects by inducing expression of cytokines such as IL-10 and inhibiting expression of TNFα, which induces insulin resistance and mitochondrial dysfunction in myocytes in vitro. Similarly, irisin may increase glucose transporter 4 (GLUT4) expression and mitochondrial uncoupling and biogenesis in cultured myocytes [64].

Both obesity and T2D influence myokine profiles; however, the results of the studies regarding this issue are inconsistent. HFD decreases, e.g., IL-6 secretion in cultured rat and human myocytes [65,66]. However, obese subjects with impaired glucose tolerance or T2D have increased expression of IL-6, TNFα, and MCP-1 in SM compared with healthy controls [67]. Nevertheless, changes in myokines secretion do not constitute the major component of SM inflammation in T2D and obesity, since the most characteristic feature of this process is an increased muscle infiltration by macrophages and T cells [68]. This hypothesis is supported by the finding that even a short-term high-fat, high-calorie diet, or overfeeding leads to increased infiltration of SM with macrophages and parallel induction of insulin resistance in healthy subjects [69].

Histologically, macrophages and T cells are primarily located in the intermuscular (IMAT) or perimuscular (PMAT) adipose tissues that expand in the course of obesity, and their volume correlates with insulin resistance [70]. As in the case of the adipose tissue and the liver, as a consequence of overnutrition, immune cells infiltrating SM polarize towards a pro-inflammatory phenotype that is reflected by an increased number of M1 macrophages and Th1 cells and decreased number of Treg lymphocytes and was observed in mice on HFD and in the obese individuals [70]. These phenomena are followed by the substantial increase in the secretion of pro-inflammatory cytokines by the cells infiltrating IMAT and PMAT, significantly higher than those that occur in the myocytes reviewed in [64].

To sum up, both animal and human studies support the hypothesis that obesity is associated with increased inflammation in SM. Although myocytes contribute to the inflammatory state, the majority of pro-inflammatory mediators are secreted by the immune cells infiltrating IMAT and PMAT. These cells are attracted to the muscles by the chemokines secreted by myocytes and adipocytes overloaded with FFA [71].

### 2.4. Insulitis

Even though at the first stages of T2D natural history, the pancreas increases insulin secretion in response to insulin resistance, it does not mean that β cells are functioning normally. The onset of β cell failure occurs earlier and is more severe than it was previously considered, and the impaired glucose tolerance (IGT) occurs where pancreas losses from 60% to 85% of the total insulin secretory capacity. The final progression of the obese, prediabetic individual to overt diabetes is synonymous with a decline in insulin secretion without any worsening of insulin resistance [4]. For years insulitis has been considered as a chief pathomechanism of type 1 diabetes (T1D); however, it occurred to be the main reason for the pancreatic islets failure in T2D, too [72].

Pancreatic islets of T2D patients are characterized by the infiltration of the immune cells accompanied by the deposition of amyloid reviewed in [73]. What differs insulitis in T1D from T2D is a composition of the immune cell infiltration: in T1D, it consists of mainly T cells, while in T2D of macrophages [74]. This observation from human studies was confirmed in several rodent models of T2D that supported the link between the immune cell infiltration with β cell dysfunction and the loss of β cell mass. Moreover, insulin resistance-induced hyperinsulinemia enhances the pro-inflammatory M1-polarization of intrapancreatic macrophages [75]. However, it is still unclear if the macrophages infiltrating pancreatic islets are recruited from the circulation or proliferate from the islet resident population reviewed in [73].

The key pro-inflammatory cytokine involved in human β cell dysfunction seems to be IL-1β that activates expression of downstream cytokines and chemokines, namely IL-6, IL-8, TNFα, chemokine (C-X-C motif) ligand 1 (CXCL1) and C-C Motif Chemokine Ligand 2 (CCL2) [76]. Moreover, studies on animals with a β cell-specific knockout of IL-1 receptor (IL-1Ra) proved that the deleterious effects of IL-1β on β cell function and islet size do not result only from its pro-inflammatory properties but also from its direct impact on β cells [77]. Subsequently, application of a recombinant human IL-1Ra (anakinra) was found to effectively reduce the rate of insulitis in animal models of T2D and patients, that was reflected by the increased insulin to proinsulin plasma ratio, improvement of blood glucose control and peripheral insulin sensitivity (see below) [78]. The role of other cytokines in the development of T2D associated insulitis is less evident. Transcriptome studies performed in islet specimens from T2D patients provided conflicting results. While some authors reported no difference in the pro-inflammatory gene expressions, others found that pancreatic islets of T2D patients are characterized by the enrichment of IL-1-related genes (e.g., IL-6, IL-11, IL-24, IL-33) that was associated with impairment of insulin secretion reviewed in [73].

Both high serum glucose and saturated fatty acid levels have been reported to trigger the islets inflammation directly (e.g., by causing mitochondrial dysfunction, increasing ER stress, targeting toll-like receptors or the renin-angiotensin system), or via glucose-induced upregulation of the islet amyloid system in several in vitro models reviewed in [73,79]. The islet amyloid polypeptide (IAPP, also called amylin) is one of the most potent inducers of IL-1β expression and promotes chemokines secretion by not-immune cells. IAPP is co-secreted with insulin, and in the course of insulin resistance-induced hyperinsulinemia, its secretion proportionally increases. Amyloid deposits are present in β cells of most T2D patients and correlate with β cell apoptosis [80].

## 3. Anti-Inflammatory Strategies Targeting Metaflammation in Type 2 Diabetes

In general, pharmacological treatment of T2D is mainly focused on glycemic control and, in this way, on the reduction of hyperglycemia-associated damage, dysfunction, and failure of various organs. Some of the available drugs reduce the risk of diabetes-associated complications in the mechanisms other than those associated with lowering glycemia, while others can even delay the onset of diabetes by slowing the progressive decline in insulin secretion. Compounds acting in all these directions would constitute an ideal treatment for patients with T2D.

Given the critical role of inflammation in the development of both: insulin resistance and insulitis, anti-inflammatory strategies could be an option not only to cure the symptoms but also to stop the vicious cycle of metaflammation. The available, non-invasive therapeutic options are summarized in Table 1 and Table 2.

### 3.1. Lifestyle Intervention to Reduce Metaflammation

#### 3.1.1. Diet and Natural Compounds

Western dietary patterns, including high consumption of sugars and saturated fats and a low intake of fruits, vegetables, and fiber consecutively correlate with higher levels of inflammatory parameters as it was shown in cross-sectional studies [81]. Since high-fat and high-calorie diets induce metaflammation, the idea that dietary intervention might be helpful to reduce the inflammatory response in T2D is plausible. Indeed, the efficiency of different dietary protocols in the reduction of systemic inflammation has been reported in clinical studies reviewed in detail in [82]. These beneficial effects were observed not only in case of low-calorie, low-fat (≤30% of energy from fat) diets but also for the Dietary Approaches to Stop Hypertension (DASH) protocol as well as for high-protein and low-carbohydrate diets [83,84,85,86,87]. Even though the studies have not been homogenous in their design, all reported that weight reduction, improvement in glycaemic control and degree of liver steatosis, were associated with a decrease in serum CRP levels in varying degrees.

The potential mechanism underlying these phenomena is associated with the beneficial effects of a decrease in total energy, total fat, and carbohydrate intake on tissue metabolism that is reflected by the reduced inflammatory response. However, we can infer this indirectly since the intensity of inflammatory infiltration in tissues has been evaluated in none of these studies. What is also worth mentioning, the benefits in the inflammatory status could be contributed to the weight loss rather than the specific elimination of pro-inflammatory compounds and/or inclusion of compounds with anti-inflammatory properties shown in vitro [88]. Moreover, most clinical studies on the influence of the dietary intervention on the course of metaflammation have several limitations that include: (i) small sample sizes, limiting the power of the intervention to detect an effect; (ii) lack of a transparent definition of the dietary intervention, rendering it difficult to evaluate the role of a particular dietary modification on the inflammatory status; (iii) difficulties in assessing compliance with dietary recommendations.

One of the best-studied in the context of metaflammation is the Mediterranean diet (MD). In a randomized controlled trial (RCT), adherence to the Mediterranean diet decreased not only CRP levels but also TNFα and IL-6 concentrations, parallel to the improvement of glycemic control in T2D patients [89]. Moreover, MD reduces the incidence of T2DM irrespectively of BMI since it is not a calorically-restricting [90]. It is therefore plausible that anti-inflammatory effects of MD are related to its composition, including functional foods containing polyphenols, terpenoids, flavonoids, alkaloids, sterols, pigments, unsaturated fatty acids and others [91]. It is not possible to attribute inflammation and T2DM risk-reduction benefits to a single functional food or a nutraceutical in MD, however, the anti-inflammatory potential of some of its components have been examined in preclinical and clinical trials. For instance, supplementation with monounsaturated fatty acids (MUFA) alone for three months resulted in T2D patients in a significant reduction of CRP and IL-6 serum levels, comparable to this achieved by exercise and exercise combined with increased MUFA intake [92]. Similar findings concern increased consumption of foods reach in α and β-carotenes [93]. Moreover, supplementation with carotenoids such as cryptoxanthin and astaxanthin, that exhibit antioxidant and anti-inflammatory effects, and favorable regulate M1/M2 macrophage polarization in the liver, occurred to be effective in prevention and reversal of lipotoxicity-induced hepatic insulin resistance and steatohepatitis in mice. However, there is no evidence that carotenoids exhibit beneficial effects in patients with NAFLD reviewed in [56].

Another representative of the MD compound with the potential to combat metaflammation is resveratrol (3,5,4′-trihydroxy-trans-stilbene, RSV). Pleiotropic effects of RSV on human organisms include, among others, antioxidant and anti-inflammatory activities. These are predominantly exerted by direct modulation of NF-κB activation or by remodeling of chromatin through regulation of histone deacetylase (as sirtuins) activity and subsequently by down-regulation of inflammatory gene expression. Exposure of human adipose tissue explants and differentiated preadipocytes in primary culture to RSV effectively reversed the increased expression of pro-inflammatory cytokines (IL-6, IL-8, MCP-1) caused by exposure to IL-1β [94]. Its administration to animals can reverse detrimental effects of HFD on adipose tissue content, liver, muscle, and pancreatic islets steatosis as well as an inflammatory profile that subsequently results in increased insulin sensitivity decreased fasting blood glucose and insulin levels [95,96,97]. These changes were accompanied by the increased number of Tregs in the circulation [96]. RSV was also demonstrated to reduce oxidative damage in β cells of type 2 diabetic animals (*db*/*db* mice) that resulted in improved islet structure and function [98]. Finally, a meta-analysis of clinical trials revealed that in T2D patients, daily supplementation with RSV ≥ 100 mg significantly improved the fasting plasma glucose and insulin levels, homeostasis model assessment of insulin resistance (HOMA-IR) index [99].

Apart from RSV, other dietary phytoestrogens that can be found in MD (e.g., isoflavones: genistein, daidzein, and glyctin) via improvement of serum lipid profile or liver steatosis occurred to increase insulin sensitivity and lower plasma glucose and insulin levels in different animal models of nongenetic T2D reviewed in [100]. Moreover, these isoflavones can stabilize β cell function and postpone the onset of diabetes in non-obese diabetic (NOD) and streptozotocin (STZ)-induced diabetic mice [101,102]. Also, cross-sectional studies and clinical trials suggest a favorable influence of dietary isoflavones on glucose metabolism (assessed by fasting glucose, insulin, and HOMA-IR) and T2D risk [103,104]. The mechanisms of these actions are complex but include, among others, downregulation of the NF-κB-regulated inflammatory pathways [100].

Several other dietary compounds have been tested for their utility in the treatment of metaflammation in pre-clinical studies, and listing them all is beyond the scope of this work. However, it should be underlined that the promising results of the pre-clinical studies have to be verified in clinical trials to provide the evidence base for modifying clinical practice guidelines in medical nutrition therapy for patients with T2D [82].

#### 3.1.2. Physical Activity

There is evidence that exercise can both cause and attenuate inflammation. Acute, unaccustomed exercise can cause muscle and connective tissue damage and infiltration by inflammatory cells. However, if exercise is moderate and done regularly as the tissue adapts, the physical activity reduces not only adipose tissue mass but also the ongoing inflammatory process reviewed in [105]. Moreover, exercises reduce the amount of IMAT/PMAT and improve its secretory profile (reduce TNFα production and increase anti-inflammatory cytokines secretion) with significant benefits on muscle insulin/glucose metabolism [106]. These actions can be mediated by the exercise-induced downregulation of the toll-like receptor 4 ligation of which activates pro-inflammatory cascades (e.g., NF-κB pathway)[107]. Other anti-inflammatory mechanisms triggered by exercise include (i) increase of a vagal tone which in the cholinergic anti-inflammatory reflex could lead to reductions in systemic inflammation; (ii) release of cortisol due to the activation of the hypothalamic-pituitary-adrenal axis; (iii) activation of the sympathetic nervous system and subsequent inhibition of pro-inflammatory mediators synthesis by adrenaline reviewed in detail in [105].

Single clinical studies performed in T2D patients suggest that the combined exercise seem to have more significant anti-inflammatory effects than aerobic or resistance exercise alone causing a more significant decrease in CRP, IL-6, IL-1β, TNFα, leptin, and resistin levels and a higher increase in anti-inflammatory cytokines such as IL-4, IL-10, and adiponectin [108]. However, a recent meta-analysis of eleven studies did not prove that aerobic or resistance exercise improves systemic levels of inflammatory markers in patients with T2D [109].

#### 3.1.3. Gut Microbiota

The gut microbiota (GM) seems to play a significant role in the development of metaflammation. Several studies demonstrated that a non-negligible proportion of obese subjects exhibit GM dysbiosis, which is characterized by decreased microbial gene richness and a switch in bacterial composition with an increase of species with pro-inflammatory properties reviewed in [110]. Moreover, HFD induces in mice an increased intestinal permeability and subsequent translocation of bacteria into the systemic compartment, which is associated with increased circulating lipopolysaccharide (LPS) levels and inflammatory infiltration in adipose tissue [111]. In turn, in obese individuals, lipid challenge increases intestinal permeability and associates with increased systemic inflammation and risk of T2D [112]. Shifts in the GM composition contribute to the improvement of glycaemic control and remission of T2D described in obese patients post-bariatric surgery [110]. A recent meta-analysis pointed two genera, Escherichia and Akkermansia, to be increased in GM post-bariatric surgery. The abundance of Akkermansia muciniphila is associated with insulin sensitivity and decreased inflammation in obese subjects [113]. In turn, the improvement of systemic inflammation observed in T2D patients after bariatric surgery is associated with abundance Faecalibacterium prausnitzii [114]. However, how these changes in the GM composition contribute to the improved glucose metabolism is unknown. In general, obesity and diabetes-related dysbiosis of GM can partially improve after bariatric surgery, however not to the same degree as metabolic parameters, suggesting that this effect of the surgery does not have a pivotal meaning in the reduction of metaflammation and remission of diabetes.

### 3.2. Pharmacological Treatment

#### 3.2.1. The Anti-Inflammatory Potential of Classical Antidiabetic Treatments

Several antidiabetic drugs, apart from controlling glycemia, exert anti-inflammatory effects that might be mediated via their hypoglycemic and hypolipidemic abilities or by direct modulation of the immune responses.

##### Metformin

Even though metformin constitutes a primary treatment for T2D, its molecular mechanisms of action are not fully understood. It is generally accepted that metformin via phosphorylation and activation of AMP-activated protein kinase (AMPK), increases oxidation of FFA in the liver and in the SM that reduces lipotoxicity and improves insulin sensitivity. Activation of AMPK also leads to increased nitric oxide (NO) synthesis, while inhibition of nicotinamide adenine dinucleotide phosphate (NADPH) oxidase and the respiratory mitochondrial chain, decreases the production of reactive oxygen species (ROS) reviewed in [115]. In vitro and animal studies suggest that metformin also exerts several anti-inflammatory effects. These include, among others, inhibition of NF-κB cascade through both: activation of AMPK and blockade of the phosphoinositide 3-kinase (PI3K)-Akt pathway; and inhibition of advanced glycation end-products (AGEs) synthesis reviewed in [116].

In an in vitro study, treatment with metformin reduced production of NO, prostaglandins, and pro-inflammatory cytokines (IL-1β, IL-6, and TNFα) and increased synthesis of anti-inflammatory IL-4 and IL-10, in murine, stimulated with LPS, macrophages [117]. In animal models of HFD-induced obesity treatment with metformin led to the downregulation of TNFα levels and an increase in Tregs number, parallel to the improvement of adipose tissue, muscle, and liver histology [115,118].

However, data coming from clinical studies regarding the anti-inflammatory properties of metformin is inconsistent. On the one hand, the administration of metformin reduced serum levels of CRP and monocyte release of TNFα, IL-1β, IL-6, MCP-1, and IL-8 in pre-diabetic patients [106,119]. On the other, in the LANCET Trial metformin, had no significant influence on inflammatory parameters (CRP, IL-6, soluble TNF receptor 2) in patients with a short T2D duration [120].

In summary, despite the promising results of the pre-clinical experiments, the anti-inflammatory properties of metformin have not been explicitly confirmed in humans. Moreover, it is not clear whether the anti-inflammatory actions of metformin are not only secondary to the improvement of insulin sensitivity, hyperglycemia, and atherogenic dyslipidemia (reviewed in 112). Treatment with metformin is also known to increase the abundance of Akkermansia in the intestine, suggesting that some of the anti-inflammatory effects of metformin might be expressed via its influence on the microbiome [110].

##### Sulphonylureas

Some of the sulphonylureas exert an anti-inflammatory potential that was observed *in vitro*, as well as in human studies. For instance, glyburide via inhibition of the NLRP3 inflammasome (NOD-, LRR- and pyrin domain-containing protein 3, a critical component of the innate immune system) decreases IL-1β secretion in murine and human macrophages [121]. In the same mechanism, glibenclamide reduces macrophage infiltration in the heart and production of IL-1β and TNFα in cardiomyocytes in an LPS-induced myocardial injury in STZ diabetic mice [122]. In human studies, glibenclamide was found to reduce IL-1β synthesis by neutrophils in T2D patients in response to bacterial infection [123].

Another anti-inflammatory mechanism of sulphonylureas action is related to their ability to block the voltage-dependent K^+^ channels, which have a crucial role in T cells activation. In this way, chlorpropamide was found to inhibit IL-2 production and lectin-induced T cells in vitro as well as in a clinical trial in T2D patients [124]. In turn, gliclazide via interaction with PI3K cascade significantly reduced serum levels of adhesion molecules and high sensitive CRP in diabetic individuals [125].

However, when compared to other anti-diabetic therapies (pioglitazone, glucagon-like peptide 1 receptor agonists, and metformin), the use of sulphonylureas did not have an advantage in terms of reducing the concentration of inflammatory parameters (namely IL-1, IL-6, and TNFα) sulphonylureas reviewed in [126,127].

##### Thiazolidinediones

After the withdrawal of rosiglitazone from the market, the only available representative of this group of antidiabetic compounds is pioglitazone. Pioglitazone is a selective agonist of peroxisome proliferator-activated receptor γ (PPARγ), which acts as a transcription factor crucial for adipogenesis but also for the silencing of the inflammatory response in adipose tissue, steatotic liver, atherosclerotic plaques and plasma reviewed in [128].

Pioglitazone inhibits MCP-1 and induces adiponectin secretion in murine preadipocytes and causes favorable changes in adipose tissue in HFD mice by reducing the density of crown-like structures (CLS) consisting of dead or dying adipocytes surrounded by macrophages and suppressing the synthesis of pro-inflammatory mediators (e.g., TNFα, tumor growth factor β (TGFβ), and MCP-1) [129].

However, the relation between the pioglitazone treatment and metabolic outcomes is not straightforward since its insulin-sensitizing and anti-inflammatory effects occur in the presence of an increase in body weight and whole-body adiposity [130]. These conflicting findings result from the ability of pioglitazone to improve the quality of the adipose tissue, which is manifested by the increase in the number of small adipocytes with high lipid storage capacity. This phenomenon translates to the lower FFA serum levels, decrease in liver and muscle steatosis, and subsequently—improvement in inflammatory parameters and insulin resistance reviewed in [128]. In diabetic obese patients, treatment with pioglitazone decreased insulin resistance by reduction of inflammatory infiltration in adipose tissue and the liver. Moreover, pioglitazone led to the decrease in hepatic M1 and an increase in the M2 macrophages population what correlated with a reduction of inflammatory parameters (CRP, A-1-acid glycoprotein, serum amyloid A) in circulation [131,132]. Interestingly, the anti-inflammatory properties of pioglitazone are independent of its influence on glucose metabolism, since they take place in diabetic patients even though they do not notice a significant improvement of glycaemic control [133].

##### SGLT-2 Inhibitors

Sodium-glucose cotransporter 2 inhibitors (SGLT-2i) lower plasma glucose levels by inhibiting renal glucose reabsorption, but also demonstrate several benefits related to cardiovascular and renal risk factors [134]. These actions can be partially contributed to the anti-inflammatory potential of SGLT-2i that has been indicated in vitro and animal models.

In T2D animal models, SGLT-2i (namely empagliflozin, dapagliflozin, ipragliflozin, and luseogliflozin), apart from improvement of glucose, insulin and lipids levels, significantly reduced serum concentrations of inflammatory parameters (e.g., IL-1β, IL-6, TNFα, MCP-1, ICAM-1 and CRP) reviewed in [135]. Moreover, empagliflozin treatment of diabetic mice led to the reduction of M1 macrophage infiltration in adipose tissue and liver and induction of the anti-inflammatory M2 population. The molecular mechanisms of these phenomena are not fully understood. However, they may partially resemble metformin action: SGLT-2i were found to activate AMPK in vitro (directly, or by increasing adiponectin expression) that results in increased fatty acid oxidation and decreases lipid accumulation in adipose tissue and other localizations (e.g., in the liver and SM) [136]. SGLT-2i can also improve mitochondrial function by restoring ROS and ATP production in vitro [137].

Evidence for anti-inflammatory effects of SGLT-2i from clinical trials in human subjects is limited. In some studies, treatment with empagliflozin and canagliflozin reduced interferon-λ, TNFα, and IL-6 serum levels, while with dapagliflozin—also CRP concentration with a parallel increase of adiponectin in T2D patients reviewed in [135]. However, in another trial, canagliflozin did not influence adiponectin, IL-6, TNF-a, and CRP levels [138]. Given the relatively short time of clinical experience with SGLT-2i, further prospective and sufficiently powered studies are required to evaluate their potential to counteract metaflammation in humans.

##### Incretin-Based Drugs

The main mode of action of glucagon-like peptide 1 receptor agonists (GLP-1 RA) is an activation of the GLP-1 receptor on the pancreatic β cells with a subsequent increase in the intracellular concentration of cyclic adenosine monophosphate (cAMP) resulting in insulin secretion. However, these compounds possess anti-inflammatory properties, which are independent of their beneficial influence on glycaemic control and body weight.

Treatment of cultured human islets with exendin-4 inhibits activation of the NF-κB pathway that results in the down-regulation of inflammatory genes, suggesting that GLP-1 RA therapy can reduce insulitis and promote β cell survival [139]. Anti-inflammatory effects of GLP-1 RA were shown on several animal models of T2D and obesity and above all concerns reduction of macrophage infiltration in adipose tissue, liver, and SM that associated with the decreased synthesis of inflammatory mediators and increase of insulin sensitivity reviewed in [140]. In clinical trials, the treatment of diabetic patients with GLP-1 RA also leads to reduced pro-inflammatory activation of macrophages and favorable changes in the serum cytokines profile (decrease of TNFα, IL-1β, and IL-6 and increase of adiponectin level) [141].

Dipeptidyl peptidase 4 inhibitors (DPP-4i) are the other group of incretin-based drugs routinely used for T2D treatment. By inhibition of DPP-4 activity, they prevent the inactivation of the natural incretins (e.g., GLP-1) and, in this way, potentiate their influence on β cells. However, due to the ubiquitous tissue expression of DPP-4, these compounds also exert extra-glycemic effects, including immunomodulatory. Treatment of human macrophages with the DPP-4i sitagliptin resulted in the increased synthesis of cAMP in the cytosol, which prevents NF-κB nuclear translocation [142]. In turn, when administered to obese insulin-resistant mice, sitagliptin, apart from the improvement of metabolic parameters, reduced inflammatory infiltration in adipose tissue and pancreatic islets [143]. Finally, in clinical trials, treatment of T2D patients with DPP-4i effectively reduced the expression of inflammatory cytokines and favorably modified macrophage subpopulations in peripheral blood reviewed in [144].

##### Insulin

*In vitro* and animal studies suggest that insulin can exert the anti-inflammatory properties via triggering different molecular mechanisms that include, among others, stimulation of NO synthesis, activation of PI3K cascade, inhibition of toll-like receptors activation, and their downstream pathways (e.g., NF-κB and Egr-1) [145].

Administration of insulin, via inhibition of NF-κB, ameliorated the endotoxin-induced systemic inflammatory response in rats by decreasing pro-inflammatory signal transcription factors and cytokine (TNFα, IL-1β, IL-6) expression in the liver and serum, and increasing levels of anti-inflammatory mediators (IL-2, IL-4, and IL-10) [146]. The same mechanism was triggered by insulin infusion in obese, non-diabetic subjects and led to the decrease in ROS generation by peripheral blood mononuclear cells, as well as serum levels of MCP-1, plasminogen activator inhibitor-1 (PAI-1) and soluble ICAM [147]. The modulation of T cell differentiation, promoting a shift toward a Th2-type anti-inflammatory response may also contribute to the favorable influence of insulin on chronic inflammation associated with obesity and T2D [148]. In turn, activation of the PI3K cascade seems to be a chief mechanism responsible for the anti-apoptotic effect of insulin on macrophages that is associated with the improvement of their phagocytosis and oxidative burst capacity in experimental animals reviewed in [149].

The results of clinical studies evaluating the effect of insulin treatment on the course of metaflammation in T2D patients are conflicting. In some RCTs, the administration of insulin to diabetic patients decreased serum levels of CRP, IL-6, more effectively than metformin. At the same time, other studies have not shown an advantage of insulin over metformin in this aspect or even found that individuals with insulin-induced weight gain are characterized by the increased macrophages infiltration of adipose tissue with subsequent increased expression of pro-inflammatory cytokines blood reviewed in [144].

In summary, most of the available antidiabetic agents, apart from their glucose-lowering attributes, exert immunomodulatory effects that include reduction of the inflammatory infiltration in peripheral tissues and pancreatic islets with a subsequent decrease in the synthesis of the pro-inflammatory mediators. Most of these actions have been shown in in vitro studies and animal models of diabetes, but translate to clinical practice. In many cases, however, it remains unclear to what extent the immunomodulatory effect of antidiabetic drugs results from their effect on glycemia, lipid levels, and body weight, and to what extent is the result of the activation of other mechanisms.

#### 3.2.2. Anti-Inflammatory Strategies in the Treatment of Metaflammation

##### Classic Nonsteroidal Anti-Inflammatory Drugs (NSAIDs)

NSAIDs represent a heterogeneous group of compounds with differently expressed anti-inflammatory, antipyretic, and analgesic properties that result from the inhibition of cyclooxygenases (COX). Among NSAIDs, aspirin is the only one that, by acetylation of COX-1 and COX2, irreversibly inhibits the synthesis of prostaglandins. Other NSAIDs interact with COX enzymes by competitive antagonism and with different selectivity. Given the role of inflammatory processes in the development and progression of T2D, the idea of application of NSAIDs to counteract metfalmmation is plausible. However, the efficacy of NSAIDs on reducing the progression of T2D is still far from being demonstrated, and the use of these drugs is not free of side effects.

Salicylates represent the most commonly used nonsteroidal anti-inflammatory drugs. The first announcements concerning the ability of salicylates to improve glucose metabolism came at the end of the XIX century. This effect was largely forgotten until the emergence of recent studies linking inflammation with the development of obesity and T2D. The hypoglycemic actions of salicylates have been reinvestigated in animal models, and members of the NF-κB cascade seem to be their primary molecular targets [150].

Several relatively small, clinical studies reported that a short-term (up to 3 weeks) high-dose (up to 10 g/day) aspirin treatment improves glucose tolerance and may ameliorate insulin resistance in diabetic patients [151]. Besides, high-dose aspirin treatment had a beneficial influence on total cholesterol and triglycerides serum concentrations. However, the therapeutic potential of high-dose aspirin is limited by bleeding risk (mainly in the gastrointestinal tract). Low-dose aspirin therapy (30–100 mg) was found to be also useful in triggering the anti-inflammatory mediators’ production and lowering systemic levels of inflammatory biomarkers [152]. However, in an RCT, long-term treatment with a low-dose aspirin occurred to be ineffective in the prevention of T2D [153]. In this study, 38,716 participants (women aged ≥ 45, free of clinical diabetes, enrolled in the Women’s Health Study) had been randomly assigned to either low-dose (100 mg/day) aspirin or placebo and followed-up for ten years. Although the two groups were similar in the number of participants (19,326 vs. 19,390) and the adherence, the incidence of T2D was almost equal (849 cases in the aspirin vs. 847 in the placebo group). Additional stratification by diabetes risk factors including age, BMI, positive family history, physical activity, glycated hemoglobin (HbA1C), and CRP concentrations did not support any modulating effect of these variables.

Salsalate, a prodrug of salicylate that reversibly inhibits COX, has also been shown to improve glycemic control both in diabetic and nondiabetic subjects. One month of treatment with 3.5 g/d salsalate (versus placebo) reduced fasting glucose and CRP levels, while increased circulating adiponectin concentrations in obese adults [154]. Three weeks of treatment with salsalate occurred to be effective in reducing fasting glucose and insulin resistance in prediabetic individuals without causing liver or kidney complications [155]. In turn, one-year-long administration of this compound to T2D patients led to a significant decrease in fasting glucose and HbA1c level, however, with a higher tendency to mild hypoglycemic events, especially in individuals receiving sulfonylurea and without improvement in vascular inflammation [156].

The usefulness of salicylates in T2D was assessed in a meta-analysis involving 13,464 patients, that revealed that while any dose of salicylates could significantly reduce HbA1c level, only high doses (≥3000 mg/day) could effectively reduce fasting plasma glucose and triglycerides with a simultaneous increase in plasma fasting insulin level with a relatively low risk of adverse effects [157]. The influence of salsates on insulin concentration results rather from its reduced clearance than the improvement of β cell function reviewed in [158]. In turn, the glucose-lowering and anti-inflammatory properties of salsates do not seem to result from the reduction of the inflammatory infiltration and cytokine synthesis by adipose tissue, but rather from the reduced liver inflammation. This hypothesis is supported by a clinical trial where salsates, by inhibition of IκBs (kinases acting as inhibitors of NF-κB), effectively decreased hepatic insulin resistance caused by high FFA plasma concentrations [159].

Animal studies suggest that other NSAIDs can present favorable properties in the context of metaflammation, too. For instance, indomethacin, combined with fish oil, significantly reduced dyslipidemia and liver steatosis in LDLR_/_mice (a model of familial hypercholesterolemia) [160]. In turn, celecoxib (selective COX-2 inhibitor) reversed NASH in rats with diabetes induced by a high-fat and sucrose (HF-HS) diet that was reflected by a reduced inflammatory infiltration and lower activity of pro-inflammatory pathways in the livers [161]. However, the anti-diabetic potential of indomethacin has to be evaluated in clinical trials.

In summary, animal studies suggest that the potential effects of NSAIDs on glucose homeostasis include the increase of insulin sensitivity via reduction of systemic inflammation and improvement of insulin secretion due to the anti-inflammatory effect on pancreatic islet and reduction of the insulin clearance. Results of clinical studies concern mainly salicylates are promising; however, they did not allow to create clear guidelines regarding the recommendation of these compounds in T2D prevention or treatment. In turn, the application of other NSAIDs in order to combat metaflammation requires clinical trials.

##### Anticytokines

Since imbalance between pro- and anti-pro-inflammatory cytokines plays a central role at every stage of development and progression of metaflammation, the anti-cytokine therapies could constitute an alternative for its treatment. Subsequently, different anticytokines have been evaluated for their influence on glucose level, insulin resistance, and pancreatic islets function.

The favorable effects of anti-TNFα therapy on insulin resistance and risk of T2D were initially observed in patients with rheumatoid arthritis, psoriasis, ankylosing spondylitis, and inflammatory bowel diseases where compounds neutralizing TNFα action (such as infliximab or etanercept) were used to decrease inflammation and slow down the course of the autoimmune disease reviewed in [162]. On the molecular level, the increase in insulin sensitivity during anti-TNFα therapy is related to the restoration of the insulin signaling cascade [163]. Next, TNFα inhibition was found to significantly decrease fasting glucose in obese, non-diabetic individuals that was associated with the increase of total adiponectin concentration, but with an increase in muscle adiposity [164,165]. These contradictory findings require verification in prospective clinical trials to verify if prolonged TNFα antagonism can have a beneficial influence on T2D.

Given the critical role of IL-1β in the development of β cell dysfunction, the influence of IL-1β neutralization on glucose tolerance was tested in several animal models of T2D. Treatment with the IL-1R antagonist (IL-1Ra) resulted in the decreased infiltration of immune cells in islets of GK rats (a spontaneous, non-obese model of T2D), that resulted in improvement of insulin secretion and glycaemic control [166]. Improvement of β cell function and reduction of insulitis after treatment with IL-1Ra was also observed in a rodent model of islet amyloidosis [167]. In turn, specific anti-IL-1β antibody diminished islet infiltration, β cell apoptosis, and led to improved insulin secretion and blood glucose control in mice on HFD [168]. Further, a recombinant human IL-1Ra (anakinra) partially reversed β cell dysfunction in human islet cultures induced by gluco- and lipotoxicity [76,169].

Anti- IL-1β therapies have also been tested for their effectiveness in T2D treatment in humans. A recent meta-analysis of clinical trials, which included almost three thousand T2D patients, showed that the administration of IL-1β antagonists leads to a significant reduction of HbA1c [170]. Moreover, treatment with anti-IL-1β antibody in diabetic patients significantly reduced the risk of macrovascular and microvascular diabetic complications [171,172,173].

##### Activation of Regulatory T Cells

Treg lymphocytes are a subtype of T helper cells with immunosuppressive properties, which number is significantly reduced in the inflammatory infiltrations in different tissues of obese and diabetic experimental animals [33,71,96]. However, in humans, the association between obesity, insulin resistance, and Tregs number is less evident [36,37,174]. It is also suggested that in the course of obesity, the number of natural Tregs decreases in favor of adaptive Tregs, which can partially explain the contradictory findings regarding the abundance of Tregs population in human T2D [175]. Nevertheless, the concept of modulating the development and course of T2D by increasing the Treg population was verified in animal studies. For instance: the administration of an anti-CD3 monoclonal antibody, which enhances Tregs differentiation resulted in long-term restoration of insulin resistance and glucose tolerance in mice with diet-induced obesity (DIO) [33]. Similarly, treatment with the IL-2 antibody complex, which results in an increase in Tregs number in adipose tissue, leads to the improvement of glucose tolerance and restoration of insulin sensitivity in HFD-fed mice [36]. Presumably, a similar effect can be obtained in obese, insulin-resistant patients by the treatment with long-chain n–3 polyunsaturated fatty acids (PUFAs); however, this assumption has to be verified in clinical trials [176].

### 3.3. Bariatric Surgery

Even though lifestyle interventions leading to weight loss beneficially influence the intensity of metaflammation and glycaemic control, their effects are difficult to sustain. Nowadays, bariatric surgery is not only the most effective intervention for morbid obesity treatment but also the most effective way to induce T2DM remission, even in 70% of the operated individuals. Recommendations indicate bariatric surgery as the preferred treatment for T2D in class III obese (body mass index (BMI) ≥40) subjects, with consideration of surgery for class II obese individuals (BMI 35–39.9) with inadequate control of hyperglycemia. T2D remission rates might vary depending on the surgical procedure, studied population, and definition of T2D remission [177].

The mechanisms underlying the improvement of glycaemic control after bariatric surgery are still the subject of research. However, they include the restoration of β cell function, insulin sensitivity and glucose utilization, changes in intestinal absorption, as well as in the secretory pattern and morphology of adipose tissue. These are mediated through enhancement in gut hormones release, changes in bile acids circulation, glucose transporters expression, and, as it was mentioned above—in microbiome composition reviewed in [178].

Bariatric surgery may also contribute to the remission of T2D via its beneficial influence on metaflammation. Numerous studies reported a decrease in pro-inflammatory cytokines (e.g., TNFα, IL-6, IL-8) and acute-phase protein serum levels in bariatric patients; however, to a different extent, that probably results from the discrepancies in surgical procedures and post-surgical time points reviewed in [179]. This phenomenon is associated with morphological changes in adipose tissue (reduction in adipocyte size and number of infiltrating macrophages) and muscles (reduction of IMAT and PMAT volume) [180].

Bariatric surgery can also reverse the pathological liver changes in NAFLD (reduce hepatic fat accumulation, inflammation, and fibrosis) and prevent its progression towards NASH. These effects are obtained not only through a substantial weight loss but also via a simultaneous favorable influence on lipid metabolism and inflammatory pathways involved in NAFLD development. Even though the improvement in liver histology depends on the type of surgical procedure, it translates into a reduction of insulin resistance and, therefore, contributes to the remission of T2D. However, it has to be mentioned that not in all individuals, bariatric surgery has a beneficial influence on liver structure. In a certain percentage of patients, hepatic steatosis, lobular inflammation, and fibrosis may aggravate or even develop *de novo* after surgery reviewed in [181]. Therefore, longitudinal studies focused on the potential differences in inflammation course due to differences in surgical techniques and sample timing are essential to assess the importance of reduction in inflammation on T2D remission

## 4. Conclusions

Given the role of inflammation in the development of insulin resistance and pancreatic islets dysfunction, the anti-inflammatory therapies could be an effective way to break a vicious circle of metaflammation and provide a causative, not only symptomatic treatment. Indeed, the results of the pre-clinical studies (conducted in vitro or animal models of diet-induced obesity) suggest that this approach might be successful. However, the limited number of well-designed prospective trials makes it difficult to deduce how the results obtained from animal models translate to clinical practice.

Surprisingly, the concept of targeting metaflammation in order to improve glucose control is not particularly innovative, since most of the presently available methods of T2D treatment, in a different extend, exert the anti-inflammatory properties. However, these result from triggering multiple pathways and effects are conditioned by many factors. In contrary to this approach, the anticytokines are the example of targeted anti-inflammatory therapy, efficient in the prevention and treatment of diabetes in both animals and humans. Nowadays, high costs of treatment constitute a significant limitation in the widespread use of this strategy. Therefore further studies are required to identify novel, specific molecular pathways closely related to metaflammation, which could be therapeutically targeted to treat T2D.

## Figures and Tables

**Figure 1 molecules-25-02224-f001:**
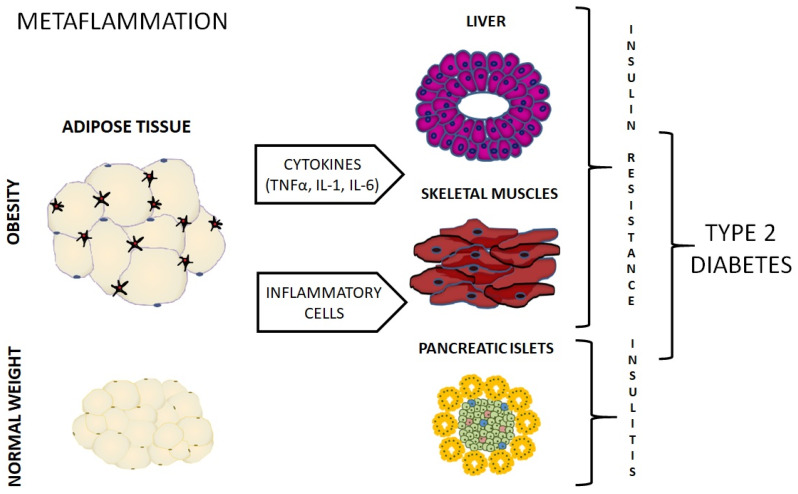
A schematic presentation of the concept of metaflammation.

**Table 1 molecules-25-02224-t001:** Summary of the life-style interventions targeting metaflammation in type 2 diabetes.

Intervention	Mechanism of Action	Influence on					Experimental Model	Ref.
		Adipose Tissue	Liver	Muscle	Pancreatic Islets	Serum/Plasma		
**Diets**								
low-calorie	↓NF-κB ↓FFA ↑ sirtuins	↓AT				↓CRP	clinical study	[82,85]
low-fat		↓AT				↓CRP	clinical study	[86]
high-protein		↓AT				↓CRP, IL-6	clinical study	[86]
high-protein low-carbohydrate		= AT				↓CRP	clinical study	[84]
DASH		↓AT	↓ALT, AST			↓CRP, TNFα, IL-6	clinical study	[83]
Mediterranean		↓= AT				↓CRP, TNFα, IL-6, IL-18	clinical study	[82,89]
**Components of the Mediterranean Diet**								
β-carotene	↓NF-κB		↓steatosis ↓M1 macrophages ↑M2 macrophages				mice on HFD	[56]
resveratrol	↓NF-κB ↑ sirtuins	↓Il-6, IL-8, MCP-1					human preadipocytes	[94]
	↓NF-κB		↓steatosis ↓TNFα, IL-6				mice on HFD	[95]
	↑ sirtuins	↓AT				↓glucose ↑ insulin, Tregs	mice on HFD	[96]
					↑β cell mass ↓islets fibrosis	↓glucose, ROS ↑ insulin	db/db mice	[98]
isoflavones	↓NF-κB ↑PPARγ ↑sirtuins		↓steatosis			↓glucose, FFA, TG	mice on HFD NOD mice	[100,101]
					↑β cell mass		human β cells	[102]
		↓VAT				↓glucose, FFA, TG	clinical trials	[104]
**Exercise**								
combined aerobic & resistance	↓NF-κB			↓IMAT ↓PMAT		↓TNFα, CRP, IL-6, IL-1β ↑IL-4, IL-10	clinical trials	[105,106,108]
**Microbiota**								
Akkermansia muciniphila						↓glucose ↓CRP	clinical trials	[113]
Faecalibacterium prausnitzii						↓glucose ↓CRP, IL-6	clinical trials	[97]

↓decrease/down-regulation, ↑increase/up-regulation, = no change, ALT—alanine transaminase, AST—aspartate transaminase, AT—adipose tissue, CRP—C-reactive protein, DASH—Dietary Approaches to Stop Hypertension diet, FFA—free fatty acids, HFD—high-fat diet, IL—interleukin, IMAT—intramuscular adipose tissue, MCP-1—monocyte chemoattractant protein-1, NF-κB—nuclear factor κB, PMAT—perimuscular adipose tissue, PPARγ—peroxisome proliferator-activated receptor γ, ROS—reactive oxygen species, TNFα—tumor necrosis factor α, TG—triglycerides, Tregs—regulatory T lymphocytes, VAT—visceral adipose tissue.

**Table 2 molecules-25-02224-t002:** Summary of the available non-invasive therapies targeting metaflammation in type 2 diabetes.

	Mechanism of Action	Influence on					Experimental Model	Ref.
		Adipose Tissue	Liver	Muscle	Pancreatic Islets	Serum/Plasma		
**Antidiabetic Drugs**								
metformin	↑AMPK ↓NF-κB ↓NO					↓ROS ↓AGEs	endothelium	[115]
						↓TNFα, IL-1β, IL-6 ↑ IL-4, IL-10	macrophages	[117]
		↓VAT ↓ROS	↓steatosis ↓ROS	↓IMAT ↓PMAT		↓TNFα ↑Tregs	rabbits on HFD	[118]
						↓TNFα, IL-1β, IL-6, MCP-1	clinical trial	[106,119]
sulphonylureas	↓ NLRP3 inflammasome ↓K+ channels ↑PI3K cascade					↓IL-1β ↓TNFα,	macrophages cardiomyocytes	[121,122]
						↓IL-2 ↓T cells activation	clinical trial	[124]
						↓CRP	clinical trial	[125]
pioglitazone	↑PPARγ					↓MCP-1 ↓adiponectin	preadipocytes	[129]
		↓TNFα, MCP-1 TGFβ					mice on HFD	[129]
		↓VAT	↓steatosis ↓M1 ↑M2 macrophages				clinical trial	[131]
						↓CRP, MCP-1	clinical trial	[132]
SGLT-2i	↑AMPK	↓AT	↓steatosis	↓IMAT ↓PMAT		↓ IL-1β, IL-6, TNFα, MCP-1, CRP	T2D mice	[136]
		↓M1 macrophages ↑M2 macrophages					mice on HFD	[136]
						↓ IFN-λ, IL-6, TNFα, CRP	clinical trial	[135]
GLP-1 RA	↓NF-κB				↑β cell survival		human β cells	[139]
		↓M1 macrophages ↑M2 macrophages ↓TNFα, IL-1β, IL-6, MCP-1					ob/ob mice	[140]
						↓IL-1β, IL-6, TNFα ↑adiponectin	clinical trial	[141]
DPP-4i	↓NF-κB					↓IL-1β, IL-6	macrophages	[142]
		↓inflammatory infiltration			↓inflammatory infiltration		T2D mice	[143]
						↓IL-6, IL-18, TNFα ↓M1 macrophages ↑M2 macrophages	clinical trial	[144]
Insulin	↓NF-κB ↑NO ↑PI3K cascade		↓TNFα, IL-1β, IL-6			↓TNFα, IL-1β, IL-6↑ IL-2, IL-4, IL-10	rats	[147]
						↓CRP, IL-6 ↓ROS,MCP-1 ↑ Th2 cells	clinical trial	[144,147]
						↑ phagocytosis	macrophages	[149]
**NSAIDs**	↓NF-κB					↓ glucose, IR, HbA1c ↓ TG ↓CRP ↑adiponectin	clinical trial	[151,154,155,156]
		↓inflammatory Infiltration ↓steatosis				↓FFA	clinical trial mice on HFD	[159,161]
**Anticytokines**								
	↓TNFα			↓IMAT↓PMAT		↓ glucose, HbA1c ↑adiponectin	clinical trial	[162,164,165]
	↓IL-1β				↓inflammatory infiltration	↓ glucose ↑insulin	T2D rats	[166]
					↓β cell apoptosis	↓ glucose ↑insulin	mice on HFD	[168]
					↑insulin		human β cells	[169]
						HbA1c	clinical trial	[170]
**Tregs**						↓ glucose, IR	DIO mice mice on HFD	[33,36]

↓decrease/down-regulation, ↑increase/up-regulation, = no change, AGEs—advanced glycation end-products, AMPK—AMP-activated protein kinase, AT—adipose tissue, CRP—C-reactive protein, DIO—diet induced, obesity, DPP-4i—dipeptidyl peptidase 4 inhibitors, FFA—free fatty acids, GLP-1 RA—glucagon-like peptide 1 receptor agonists, HbA1c—glycated haemoglobin, HFD—high fat, diet, IFN-λ—interferon λ, IL—interleukin, IMAT—intramuscular adipose tissue, IR—insulin resistance, MCP-1—monocyte chemoattractant protein-1, NF-κB—nuclear factor κB, NLRP3 inflammasome—NOD-, LRR- and pyrin domain-containing protein 3, NO—nitric oxide, NSAIDs—nonsteroidal anti-inflammatory drugs, PI3K—phosphoinositide 3-kinase, PMAT—perimuscular adipose tissue, PPARγ—peroxisome proliferator-activated receptor γ, ROS—reactive oxygen species, SGLT-2i—sodium-glucose cotransporter 2 inhibitors, T2D—type 2 diabetes, Tregs—regulatory T cells, TNFα—tumor necrosis factor α, TG—triglycerides, TGFβ—tumor growth factor β, Tregs—regulatory T lymphocytes, VAT—visceral adipose tissue.

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
