# Peer review of "Anti-Inflammatory Strategies Targeting Metaflammation in Type 2 Diabetes"

_molecules, 2020, doi:10.3390/molecules25092224_

Round 1

Reviewer 1 Report

General comment

I would like to congratulate Alina Kuryłowicz and Krzysztof Koźniewski for delivering an interesting review of literature focusing on the anti-inflammatory strategies targeting metaflammation in type 2 diabetes. While the manuscript is well-written, I have a few suggestions to improve its quality. I addition to updating citated references, briefly discussing the regulation of Th1/Th2 cytokine responses and their implication in diabetes is crucial. While enough interventions to impact the inflammatory response in diabetes are discussed, provided information is not described consistently throughout the manuscript. Specific comments are below:

Major comments:

I noticed that while most citations are accurate, but these remain relatively outdated. I suggest authors update their references, for the manuscript to remain more relevant.

While the authors are accurate in discussing specific anti-and pro-inflammatory cytokines implicated in the development of diabetes, it would be interesting if authors can briefly describe the role and modulation of T-helper type 1 (Th1) and Th2 play cytokines in terms of development of type 2 diabetes [further reading https://www.ncbi.nlm.nih.gov/pubmed/27917204 and https://www.ncbi.nlm.nih.gov/pubmed/31704479]

Briefly discuss the role of these individual cytokines and their regulation across various disease models, liver/skeletal muscle and so on… Is there a consistency in their modulation?

To accompany a brief description on adiponectin, perhaps authors can also discuss leptin and its implication in the process of inflammation in type 2 diabetic conditions.

For Figure 1, a brief description as part of the cation is necessary. Also within the figure, authors should be specific and indicate that its adipose tissue expansion they are referring to. Perhaps also indicate which cytokines are detrimental.

It is great that authors tried to cover as many interventions that can be beneficial to counteract pro-inflammation, including adding useful tables to guide the reader. However, I feel that information under some subheadings can be improved significantly. For example, consistently, enough background should be provided for each intervention. What is reported across in vitro in comparison to clinical settings for each intervention? If you are discussing dietary interventions for example, why did you choose those ones like resveratrol while leaving others? While this makes sense because you cannot discuss everything, but you should guide the reader in terms of applied search strategy or mentioning the preference of choosing those ones. Beyond stimulating NO synthesis, activation of PI3K-Akt plays a crucial role in the activity of insulin leading to amelioration of insulin resistance [ncbi.nlm.nih.gov/pmc/articles/PMC3992527/]. Overall, authors have to take time and try to make each section discussing interventions to be precise and have impact, so that all interventions have a meaning and can provide the reader with useful information.

Minor comments

For statements mentioning population based estimates, be sure to cite the precise references

Line 86-97, add relevant citations

Line 222-226, add relevant citations

Under the subheading “metformin”, add relevant citations

Spell-check and sentence construction is necessary.

Other sentences don’t have full stops.

Avoid using the same word twice in one sentence.

Author Response

Reviewer 1

Comments and Suggestions for Authors

General comment

I would like to congratulate Alina Kuryłowicz and Krzysztof Koźniewski for delivering an interesting review of literature focusing on the anti-inflammatory strategies targeting metaflammation in type 2 diabetes. While the manuscript is well-written, I have a few suggestions to improve its quality. I addition to updating citated references, briefly discussing the regulation of Th1/Th2 cytokine responses and their implication in diabetes is crucial. While enough interventions to impact the inflammatory response in diabetes are discussed, provided information is not described consistently throughout the manuscript.

We thank the Reviewer for the positive reception of our manuscript and drawing attention to some weak points we tried to improve.

Major comments:

  • I noticed that while most citations are accurate, but these remain relatively outdated. I suggest authors update their references, for the manuscript to remain more relevant.

Following the Reviewer's suggestion, we updated several citations within the manuscript, apart from a few which can be considered as "milestones" in the research on metabolic inflammation (like these, e.g., by DeFronzo et al.). All changes are marked in green in the revised version.

  • While the authors are accurate in discussing specific anti-and pro-inflammatory cytokines implicated in the development of diabetes, it would be interesting if authors can briefly describe the role and modulation of T-helper type 1 (Th1) and Th2 play cytokines in terms of development of type 2 diabetes [further reading https://www.ncbi.nlm.nih.gov/pubmed/27917204 and https://www.ncbi.nlm.nih.gov/pubmed/31704479]

We do agree with the Reviewer that the role of Th1/Th2 responses in the development of T2D required a broader discussion; therefore, we added a new paragraph to section 2.1. Inflammation of adipose tissue with proper references.

“In humans, a systemic imbalance in Th1/Th2 response has been linked with disturbances in glucose homeostasis leading to the development of insulin resistance and T2D [31]. T2D patients are characterized by the elevated concentrations of cytokines belonging to Th1 responses (such as, e.g., IFN-γ and TNFα) with a relative suppression of Th2 and Tregs-related immunosuppressive cytokines (such as, e.g., IL-4, Il-10, and IL-10, IL-35, respectively). The enhanced Th1 profile with suppression of Th2 and Tregs cytokines correlates with the intensity of oxidative stress, insulin resistance, and development of micro- and macrovascular complications such as retinopathy, nephropathy and coronary artery disease [reviewed in 32].” (lines 158-165)

31. Zhou, T.; Hu, Z.; Yang, S.; Sun, L.; Yu, Z.; Wang, G. Role of Adaptive and Innate Immunity in Type 2 Diabetes Mellitus. J Diabetes Res 2018, 2018, 7457269.

32. Mahlangu, T.; Dludla, P. V.; Nyambuya, T. M.; Mxinwa, V.; Mazibuko-Mbeje, S. E.; Cirilli, I.; Marcheggiani, F.; Tiano, L.; Louw, J.; Nkambule, B. B. A systematic review on the functional role of Th1/Th2 cytokines in type 2 diabetes and related metabolic complications. Cytokine 2020, 126, 154892.

  • Briefly discuss the role of these individual cytokines and their regulation across various disease models, liver/skeletal muscle and so on… Is there a consistency in their modulation?

Following the Reviewer's suggestion, we discussed the role of particular cytokines in the development of metaflammation in various tissues.

2.1. Inflammation of adipose tissue

“Pro-inflammatory cytokines synthesized in adipose tissue influence whole body function, but in the auto- and paracrine manner, have an impact on adipocytes themselves. For instance, both TNFα and IL-6 induce insulin resistance in rodents and block insulin action in murine (3T3-L1) adipocytes. However, Il-6 can also induce oxidation of FFA and adipose tissue browning in animal models of obesity [44]. IL-1β seems to play a central role in macrophage-adipocyte crosstalk, which impairs insulin sensitivity in adipose tissue by inhibition of insulin signal transduction. Moreover, IL-1β stimulates the production of IL-8, a potent chemoattractant involved in the adhesion of monocytes to endothelium and, in the migration of vascular smooth cells, proposed, therefore, as a mediator between obesity and atherosclerosis [reviewed in 45].” (lines 192-200)

2.2. Inflammation in the liver

“The critical role of TNFα in the development of NAFLD-related insulin resistance has been proved in animal and human studies, and its neutralization can substantially reduce hepatic steatosis in genetically obese (ob/ob) mice. HFD also leads to the increased expression of IL-1α/β in the liver, and knockout of these two cytokines protects from inflammation related to diet-induced steatosis in experimental animals. Therefore IL-1 inhibitors are being considered as a therapeutic option for NAFLD treatment in humans [reviewed in 54]. Besides, obesity-associated hypoadiponectinemia and hyperleptinemia also contribute to these phenomena by impairing FFA metabolism and promoting chronic liver inflammation [55,56].” (lines 246-254)

2.3. Inflammation in the muscle

“Similar to adipocytes, myocytes secrete several cytokines, known as myokines, including IL-6, IL-8, and TNFα but also irisin, myonectin, and myostatin. Expression of genes encoding most of the myokines is regulated by exercise and muscle contraction, and their effects on glucose and lipid metabolism, as well as on inflammation can be beneficial [reviewed in 61]. For instance, IL-6 increases glucose uptake, lipolysis, and oxidation of FFA and mediates anti-inflammatory effects by inducing expression of cytokines such as IL-10 and inhibiting expression of TNFα, which induces insulin resistance and mitochondrial dysfunction in myocytes in vitro. Similarly, irisin may increase glucose transporter 4 (GLUT4) expression and mitochondrial uncoupling and biogenesis in cultured myocytes [61].” (lines 268-276)

2.4. Insulitis

“The key pro-inflammatory cytokine involved in human β cell dysfunction seems to be IL-1β that activates expression of downstream cytokines and chemokines, namely IL-6, IL-8, TNFα, chemokine (C-X-C motif) ligand 1 (CXCL1) and C-C Motif Chemokine Ligand 2 (CCL2) [73]. Moreover, studies on animals with a β cell-specific knockout of IL-1 receptor (IL-1Ra) proved that the deleterious effects of IL-1β on β cell function and islet size do not result only from its pro-inflammatory properties but also from its direct impact on β cells [74]. Subsequently, application of a recombinant human IL-1Ra (anakinra) was found to effectively reduce the rate of insulitis in animal models of T2D and patients, that was reflected by the increased insulin to proinsulin plasma ratio, improvement of blood glucose control and peripheral insulin sensitivity (see below) [75]. The role of other cytokines in the development of T2D associated insulitis is less evident. Transcriptome studies performed in islet specimens from T2D patients provided conflicting results. While some authors reported no difference in the pro-inflammatory gene expressions, others found that pancreatic islets of T2D patients are characterized by the enrichment of IL-1-related genes (e.g., IL-6, IL-11, IL-24, IL-33) that was associated with impairment of insulin secretion [reviewed in 70].” (lines 318-331)

  • To accompany a brief description on adiponectin, perhaps authors can also discuss leptin and its implication in the process of inflammation in type 2 diabetic conditions.

Following the Reviewer's suggestion, we briefly discussed the role of leptin in the development of metaflammation.

2.1. Inflammation of adipose tissue

“On the contrary, obesity is accompanied by an increased secretion of leptin. The serum level of this adipokine serves as a gauge for energy reserves and directs the hypothalamus to adjust food intake and energy expenditure. Both obese animals and humans are characterized by resistance to leptin since its high serum levels do not translate to anorectic effects. The role of leptin and leptin resistance in the development of T2D is composed since, on the one hand, leptin can decrease insulin secretion acting directly on the pancreatic β cells and increase lipid accumulation in the liver, while on the other enhance glucose uptake and oxidation in skeletal muscle. The important mechanism linking obesity-related hyperleptinemia and insulin resistance is related to pro-inflammatory properties of leptin that involve modulation of T cells action and upregulation of multiple inflammatory cytokines (including TNFα, IL-6) [reviewed in 48].” (lines 206-2016)

  • For Figure 1, a brief description as part of the cation is necessary. Also within the figure, authors should be specific and indicate that its adipose tissue expansion they are referring to. Perhaps also indicate which cytokines are detrimental.

To facilitate the interpretation of Figure 1, we added a legend and introduced the changes suggested by the Reviewer.

Figure 1. A schematic presentation of the concept of metaflammation.

Excess of nutrients in the course of obesity impairs adipocyte metabolism leading to mitochondrial dysfunction that contributes to the endoplasmic reticulum stress, hypoxia, and cell hypertrophy. These processes result in the increased expression of genes encoding cytokines, chemokines, and adhesion molecules in adipose tissue what subsequently attracts infiltrating immune cells (macrophages and different subsets of T cells) that additionally contribute to the synthesis of pro-inflammatory cytokines. Pro-inflammatory mediators (such as, e.g., tumor necrosis factor-alpha, TNFα, and interleukins 1 & 6) impair adipose tissue function in the auto- and paracrine manner but also influence other tissues including liver, muscle, and pancreatic islets. In the liver and muscles, this process leads to the development of insulin resistance while in the pancreatic islands to disturbances in insulin secretion and apoptosis of β cells. (lines 85-95)

  • It is great that authors tried to cover as many interventions that can be beneficial to counteract pro-inflammation, including adding useful tables to guide the reader. However, I feel that information under some subheadings can be improved significantly. For example, consistently, enough background should be provided for each intervention. What is reported across in vitro in comparison to clinical settings for each intervention? If you are discussing dietary interventions for example, why did you choose those ones like resveratrol while leaving others? While this makes sense because you cannot discuss everything, but you should guide the reader in terms of applied search strategy or mentioning the preference of choosing those ones. Beyond stimulating NO synthesis, activation of PI3K-Akt plays a crucial role in the activity of insulin leading to amelioration of insulin resistance [ncbi.nlm.nih.gov/pmc/articles/PMC3992527/]. Overall, authors have to take time and try to make each section discussing interventions to be precise and have impact, so that all interventions have a meaning and can provide the reader with useful information.

We have made every effort to introduce the suggested changes into the revised version of the manuscript.

In the case of the role of dietary interventions in the treatment of metaflammation, we decided to focus on the different components of the Mediterranean diet.

“One of the best-studied in the context of metaflammation is the Mediterranean diet (MD). In a randomized controlled trial (RCT), adherence to the Mediterranean diet decreased not only CRP levels but also TNFα and IL-6 concentrations, parallel to the improvement of glycemic control in T2D patients [86]. Moreover, MD reduces the incidence of T2DM irrespectively of BMI since it is not a calorically-restricting [87]. It is therefore plausible that anti-inflammatory effects of MD are related to its composition, including functional foods containing polyphenols, terpenoids, flavonoids, alkaloids, sterols, pigments, unsaturated fatty acids and others [88]. It is not possible to attribute inflammation and T2DM risk-reduction benefits to a single functional food or a nutraceutical in MD, however, the anti-inflammatory potential of some of its components have been examined in preclinical and clinical trials. For instance, supplementation with monounsaturated fatty acids (MUFA) alone for three months resulted in T2D patients in a significant reduction of CRP and IL-6 serum levels, comparable to this achieved by exercise and exercise combined with increased MUFA intake [89]. Similar findings concern increased consumption of foods reach in α and β-carotenes [90]. Moreover, supplementation with carotenoids such as cryptoxanthin and astaxanthin, that exhibit antioxidant and anti-inflammatory effects, and favorable regulate M1/M2 macrophage polarization in the liver, occurred to be effective in prevention and reversal of lipotoxicity-induced hepatic insulin resistance and steatohepatitis in mice. However, there is no evidence that carotenoids exhibit beneficial effects in patients with NAFLD [reviewed in 53].

Another representative of the MD compound with the potential to combat metaflammation is resveratrol (3,5,4’-trihydroxy-trans-stilbene, RSV). Pleiotropic effects of RSV on human organisms include, among others, antioxidant and anti-inflammatory activities. These are predominantly exerted by direct modulation of NF-κB activation or by remodeling of chromatin through regulation of histone deacetylase (as sirtuins) activity and subsequently by down-regulation of inflammatory gene expression. Exposure of human adipose tissue explants and differentiated preadipocytes in primary culture to RSV effectively reversed the increased expression of pro-inflammatory cytokines (IL-6, IL-8, MCP-1) caused by exposure to IL-1β [91]. Its administration to animals can reverse detrimental effects of HFD on adipose tissue content, liver, muscle, and pancreatic islets steatosis as well as an inflammatory profile that subsequently results in increased insulin sensitivity decreased fasting blood glucose and insulin levels [92-94]. These changes were accompanied by the increased number of Tregs in the circulation [93]. RSV was also demonstrated to reduce oxidative damage in β cells of type 2 diabetic animals (db/db mice) that resulted in improved islet structure and function [95]. Finally, a meta-analysis of clinical trials revealed that in T2D patients, daily supplementation with RSV ≥ 100 mg significantly improved the fasting plasma glucose and insulin levels, homeostasis model assessment of insulin resistance (HOMA-IR) index [96].

Apart from RSV, other dietary phytoestrogens that can be found in MD (e.g., isoflavones: genistein, daidzein, and glyctin) via improvement of serum lipid profile or liver steatosis occurred to increase insulin sensitivity and lower plasma glucose and insulin levels in different animal models of nongenetic T2D [reviewed in 97]. Moreover, these isoflavones can stabilize β cell function and postpone the onset of diabetes in non-obese diabetic (NOD) and streptozotocin (STZ)-induced diabetic mice [98,99]. Also, cross-sectional studies and clinical trials suggest a favorable influence of dietary isoflavones on glucose metabolism (assessed by fasting glucose, insulin, and HOMA-IR) and T2D risk [100,101]. The mechanisms of these actions are complex but include, among others, downregulation of the NF-κB-regulated inflammatory pathways [97].” (lines 377-419)

We have also noted that discussing all possible dietary interventions with a potential anti-inflammatory effect is beyond the scope of this work.

“Several other dietary compounds have been tested for their utility in the treatment of metaflammation in preclinical studies, and listing them all is beyond the scope of this work. However, it should be underlined that the promising results of the preclinical studies have to be verified in clinical trials to provide the evidence base for modifying clinical practice guidelines in medical nutrition therapy for patients with T2D [79].” (lines 420-424)

We have reorganized the sections discussing different therapeutic approaches in such a way, that they briefly summarize data obtained in vitro, in vivo, and clinical trials.

Metformin

“In an in vitro study, treatment with metformin reduced production of NO, prostaglandins, and pro-inflammatory cytokines (IL-1β, IL-6, and TNFα) and increased synthesis of anti-inflammatory IL-4 and IL-10, in murine, stimulated with LPS, macrophages [114]. In animal models of HFD-induced obesity treatment with metformin led to the downregulation of TNFα levels and an increase in Tregs number, parallel to the improvement of adipose tissue, muscle, and liver histology [112,115].

However, data coming from clinical studies regarding the anti-inflammatory properties of metformin is inconsistent. On the one hand, the administration of metformin reduced serum levels of CRP and monocyte release of TNFα, IL-1β, IL-6, MCP-1, and IL-8 in pre-diabetic patients [103,116]. On the other, in the LANCET Trial metformin, had no significant influence on inflammatory parameters (CRP, IL-6, soluble TNF receptor 2) in patients with a short T2D duration [117].” (lines 488-497)

Sulphonylureas

“For instance, glyburide via inhibition of the NLRP3 inflammasome (NOD-, LRR- and pyrin domain-containing protein 3, a critical component of the innate immune system) decreases IL-1β secretion in murine and human macrophages [118]. In the same mechanism, glibenclamide reduces macrophage infiltration in the heart and production of IL-1β and TNFα in cardiomyocytes in an LPS-induced myocardial injury in STZ diabetic mice [119]. In human studies, glibenclamide was found to reduce IL-1β synthesis by neutrophils in T2D patients in response to bacterial infection [120].

Another anti-inflammatory mechanism of sulphonylureas action is related to their ability to block the voltage-dependent K+ channels, which have a crucial role in T cells activation. In this way, chlorpropamide was found to inhibit IL-2 production and lectin-induced T cells in vitro as well as in a clinical trial in T2D patients [121]. In turn, gliclazide via interaction with PI3K cascade significantly reduced serum levels of adhesion molecules and high sensitive CRP in diabetic individuals [122].” (lines 507-518)

Thiazolidinedione

“Pioglitazone inhibits MCP-1 and induces adiponectin secretion in murine preadipocytes and causes favorable changes in adipose tissue in HFD mice by reducing the density of crown-like structures (CLS) consisting of dead or dying adipocytes surrounded by macrophages and suppressing the synthesis of pro-inflammatory mediators (e.g., TNFα, tumor growth factor β (TGFβ), and MCP-1) [126].

However, the relation between the pioglitazone treatment and metabolic outcomes is not straightforward since its insulin-sensitizing and anti-inflammatory effects occur in the presence of an increase in body weight and whole-body adiposity [127]. These conflicting findings result from the ability of pioglitazone to improve the quality of the adipose tissue, which is manifested by the increase in the number of small adipocytes with high lipid storage capacity. This phenomenon translates to the lower FFA serum levels, decrease in liver and muscle steatosis, and subsequently – improvement in inflammatory parameters and insulin resistance [reviewed in 125].” (lines 529-540)

Insulin

In vitro and animal studies suggest that insulin can exert the anti-inflammatory properties via triggering different molecular mechanisms that include, among others, stimulation of NO synthesis, activation of PI3K cascade, inhibition of toll-like receptors activation, and their downstream pathways (e.g., NF-κB and Egr-1) [142].

Administration of insulin, via inhibition of NF-κB, ameliorated the endotoxin-induced systemic inflammatory response in rats by decreasing pro-inflammatory signal transcription factors and cytokine (TNFα, IL-1β, IL-6) expression in the liver and serum, and increasing levels of anti-inflammatory mediators (IL-2, IL-4, and IL-10) [143]. The same mechanism was triggered by insulin infusion in obese, non-diabetic subjects and led to the decrease in ROS generation by peripheral blood mononuclear cells, as well as serum levels of MCP-1, plasminogen activator inhibitor-1 (PAI-1) and soluble ICAM [144]. The modulation of T cell differentiation, promoting a shift toward a Th2-type anti-inflammatory response may also contribute to the favorable influence of insulin on chronic inflammation associated with obesity and T2D [145]. In turn, activation of the PI3K cascade seems to be a chief mechanism responsible for the anti-apoptotic effect of insulin on macrophages that is associated with the improvement of their phagocytosis and oxidative burst capacity in experimental animals [reviewed in 146].” (lines 596-612)

Minor comments

  • For statements mentioning population based estimates, be sure to cite the precise references

Following the Reviewer's suggestion, we have changed Reference no [3].

Iglay, K.; Hannachi, H.; Joseph Howie, P.; Xu, J.; Li, X.; Engel, S. S.; Moore, L. M.; Rajpathak, S. Prevalence and co-prevalence of comorbidities among patients with type 2 diabetes mellitus. Current medical research and opinion 2016, 32, 1243–1252

  • Line 86-97, add relevant citations

Following the Reviewer's suggestion, we have added an appropriate reference [22].

Zhang, Q.; Lenardo, M.J.; Baltimore, D. 30 Years of NF-κB: A Blossoming of Relevance to Human Pathobiology. Cell 2017, 168, 37–57.

  • Line 222-226, add relevant citations.

Following the Reviewer's suggestion, we have added a relevant reference [69].

Saisho, Y. β-cell dysfunction: Its critical role in prevention and management of type 2 diabetes. World J Diabetes 2015, 6, 109–124.

  • Under the subheading "metformin", add relevant citations.

Following the Reviewer's suggestion, we have added relevant references [112-117].

112.

Rena, G.; Hardie, D.G.; Pearson, E.R. The mechanisms of action of metformin. Diabetologia 2017, 60, 1577–1585.

113.

Saisho Y. Metformin and Inflammation: Its Potential Beyond Glucose-lowering Effect. Endocr Metab Immune Disord Drug Targets 2015, 15, 196-205.

114.

Hyun, B.; Shin, S.; Lee, A.; Lee, S.; Song, Y.; Ha, N. J.; Cho, K. H.; Kim, K. Metformin Down-regulates TNF-α Secretion via Suppression of Scavenger Receptors in Macrophages. Immune network 2013 13, 123–132.

115.

Kim, E. K.; Lee, S. H.; Jhun, J. Y.; Byun, J. K.; Jeong, J. H.; Lee, S. Y.; Kim, J. K.; Choi, J. Y.; Cho, M. L. Metformin Prevents Fatty Liver and Improves Balance of White/Brown Adipose in an Obesity Mouse Model by Inducing FGF21. Mediators of inflammation 2016, 2016, 5813030

116.

Krysiak, R.; Okopien, B. The effect of metformin on monocyte secretory function in simvastatin-treated patients with impaired fasting glucose. Metabolism 2013, 62, 39-43.

117.

Pradhan, A. D.; Everett, B. M.; Cook, N. R.; Rifai, N.; Ridker, P. M. Effects of initiating insulin and metformin on glycemic control and inflammatory biomarkers among patients with type 2 diabetes: the LANCET randomized trial. JAMA 2009, 302, 1186-1194.

  • Spell-check and sentence construction is necessary. Other sentences don’t have full stops. Avoid using the same word twice in one sentence.

We tried to avoid these errors in the revised version of the manuscript.

Reviewer 2 Report

There are some aspects in this report that may be of interest to a wide audience. However, the concept of metaflammation is misleading as it is. In fact, this is a concept that resumes "chronic metabolic inflammation". With the emphasis in metabolic. Hence, 1) metabolic pathogenesis should be extended considerably. Especially those aspects related to mitochondrial dysfunction. 2)  similarly, the concept of diabetes remission should be reviewed extensively, and 3) the best-known example, with much available information is NASH and NASH remission with bariatric surgery or lifestyle approaches, deserving its own section.

The following comments denote that either the report is redirected to sterile chronic inflammation, or information on metabolism must be abundantly provided.

Author Response

Reviewer 2

There are some aspects in this report that may be of interest to a wide audience. However, the concept of metaflammation is misleading as it is. In fact, this is a concept that resumes "chronic metabolic inflammation". With the emphasis in metabolic. Hence, 1) metabolic pathogenesis should be extended considerably. Especially those aspects related to mitochondrial dysfunction. 2)  similarly, the concept of diabetes remission should be reviewed extensively, and 3) the best-known example, with much available information is NASH and NASH remission with bariatric surgery or lifestyle approaches, deserving its own section.

The following comments denote that either the report is redirected to sterile chronic inflammation, or information on metabolism must be abundantly provided.

We thank the Reviewer for drawing attention to these evident weaknesses of our work. We would like to underline, that the presented concept of metaflammation was based on the hypothesis of Gökhan Hotamisligil (Ref 7, and www.metaflammation.org) and does not refer to the infection-related metabolic alterations, as well as the metabolic and energetic programming of the immune response which are also critical components of immunometabolism. To clarify this issue, we have introduced several changes in the revised version of the manuscript and added proper references.

Abstract:

“Given the role of this sterile, not related to infection, inflammation in the development of both: insulin resistance and insulitis, anti-inflammatory strategies could be helpful not only to control T2D symptoms but also to treat its causes.” (lines 13-16)

  1. Introduction:

“It should be underlined, that the origin of obesity-related chronic inflammation is not entirely settled. Some animal studies suggest that high-fat-diet (HFD) changes gut microbiota and gut barrier function, leading to endotoxemia, lipopolysaccharide-induced activation of immune cells, and insulin resistance in rodents [reviewed in 9]. However, several lines of evidence show that intrinsic changes in adipocytes function are crucial for the activation of inflammatory responses in adipose tissue. The infection-related metabolic changes will not be covered in this review.” (lines 52-57)

  1. Concept of metaflammation

”Even though the metabolic inflammation was first described in adipose tissue, it has to be emphasized that the obesity-related influx of immune cells occurs in many other tissues such as liver, muscle, and pancreatic islets. Moreover, the stromal components and metabolic cells, such as hepatocytes, myocytes, and β cells produce mediators (e.g., cytokines and chemotactic molecules) that influence immune cells' action. These bidirectional interactions between immune, metabolic, and stromal components of particular organs are critical in determining physiological and pathological outcomes [7].” (lines 98-104)

  • metabolic pathogenesis should be extended considerably. Especially those aspects related to mitochondrial dysfunction.

We do agree with the Reviewer that focusing on immunological aspects, we somehow neglected issues related to cellular pathomechanisms leading to metabolic complications. Therefore, in the revised version of the manuscript, we added fragments regarding these aspects, especially those related to mitochondrial dysfunction.

Section 2. Concept of metaflammation

“Excess of FFA in the circulation together with the accumulation of lipids in tissues leads to the profound changes in cell metabolism, including, among others, mitochondrial dysfunction that contributes to the endoplasmic reticulum (ER) stress, hypoxia, cell hypertrophy, death and fibrosis [reviewed in 16 & 17].” (lines: 71-74)

Subsection 2.1. Inflammation of adipose tissue

“Recent evidence suggests that mitochondria are essential for maintaining metabolic homeostasis in adipocytes, and their dysfunction in the course of obesity (reflected by downregulation of cellular pathways involved in fatty acid oxidation, ketone body production and breakdown, and the tricarboxylic acid cycle) is directly associated with the intensity of inflammation and insulin resistance [19,20]. Mitochondrial dysfunction resulting in decreased fatty acid oxidation increases triglyceride accumulation. Moreover, it may trigger cell death in adipocytes and contribute to a defective differentiation of preadipocytes to mature adipocytes and adipose tissue fibrosis [reviewed 17 & 21].” (lines:112-120)

"Mitochondrial dysfunction associated with excessive adiposity is also associated with dysregulation of adipokines secretion. This concerns, e.g., adiponectin, a protein hormone almost exclusively produced in adipocytes that has anti-inflammatory, antiatherogenic, and anti-oxidative properties [reviewed in 17].” (lines: 201-204)

Subsection 2.2. Inflammation in the liver

“Lipid accumulation in the liver results from the imbalance between the delivery of lipids and their hepatic uptake, synthesis, oxidation, and secretion, and several studies suggest that mitochondrial dysfunction plays a key role in the development of advanced NAFLD and its progression to non-alcoholic steatohepatitis (NASH) [52]." (lines: 231-234)

Subsection 2.3. Inflammation in the muscles

“The pathogenesis of insulin resistance in skeletal muscle is complex and involves intramyocellular lipids deposition, mitochondrial defects, including reduced oxidative capacity and metabolic inflexibility, as well as the influence of the adipokines secreted by the malfunctioning adipose tissue in the course of obesity [59,60].” (lines: 261-265)

Subsection 2.4 Insulitis

“Both high serum glucose and saturated fatty acid levels have been reported to trigger the islets inflammation directly (e.g., by causing mitochondrial dysfunction, increasing ER stress, targeting toll-like receptors or the renin-angiotensin system), or via glucose-induced upregulation of the islet amyloid system in several in vitro models [reviewed in 70 & 76].” (lines: 332-335)

2) similarly, the concept of diabetes remission should be reviewed extensively, and

3) the best-known example, with much available information is NASH and NASH remission with bariatric surgery or lifestyle approaches, deserving its own section.

Following the Reviewer’s suggestion in the revised version of the manuscript, we have added a new section on the impact of bariatric surgery on the intensity of obesity-related inflammation, remission of diabetes, and the course of NAFLD.

Section 3.3 Bariatric surgery

“Even though lifestyle interventions leading to weight loss beneficially influence the intensity of metaflammation and glycaemic control, their effects are difficult to sustain. Nowadays, bariatric surgery is not only the most effective intervention for morbid obesity treatment but also the most effective way to induce T2DM remission, even in 70% of the operated individuals. Recommendations indicate bariatric surgery as the preferred treatment for T2D in class III obese (body mass index (BMI) ≥40) subjects, with consideration of surgery for class II obese individuals (BMI 35–39.9) with inadequate control of hyperglycemia. T2D remission rates might vary depending on the surgical procedure, studied population, and definition of T2D remission [174].

The mechanisms underlying the improvement of glycaemic control after bariatric surgery are still the subject of research. However, they include the restoration of β cell function, insulin sensitivity and glucose utilization, changes in intestinal absorption, as well as in the secretory pattern and morphology of adipose tissue. These are mediated through enhancement in gut hormones release, changes in bile acids circulation, glucose transporters expression, and, as it was mentioned above – in microbiome composition [reviewed in 175].

Bariatric surgery may also contribute to the remission of T2D via its beneficial influence on metaflammation. Numerous studies reported a decrease in pro-inflammatory cytokines (e.g., TNFα, IL-6, IL-8) and acute-phase protein serum levels in bariatric patients; however, to a different extent, that probably results from the discrepancies in surgical procedures and post-surgical time points [reviewed in 176]. This phenomenon is associated with morphological changes in adipose tissue (reduction in adipocyte size and number of infiltrating macrophages) and muscles (reduction of IMAT and PMAT volume) [177].

Bariatric surgery can also reverse the pathological liver changes in NAFLD (reduce hepatic fat accumulation, inflammation, and fibrosis) and prevent its progression towards NASH. These effects are obtained not only through a substantial weight loss but also via a simultaneous favorable influence on lipid metabolism and inflammatory pathways involved in NAFLD development. Even though the improvement in liver histology depends on the type of surgical procedure, it translates into a reduction of insulin resistance and, therefore, contributes to the remission of T2D. However, it has to be mentioned that not in all individuals, bariatric surgery has a beneficial influence on liver structure. In a certain percentage of patients, hepatic steatosis, lobular inflammation, and fibrosis may aggravate or even develop de novo after surgery [reviewed in 178].

Therefore, longitudinal studies focused on the potential differences in inflammation course due to differences in surgical techniques and sample timing are essential to assess the importance of reduction in inflammation on T2D remission.” (lines 740-773)

We have updated several citations within the manuscript, and all changes are marked in green in the revised version.

Reviewer 3 Report

In this manuscript, author reviewed anti-inflammatory strategies targeting metaflammation. This manuscript is well described about this topic. Please consider below.

P7 line 299; α and β carotenoids [72].

→They are compound name. So, α-carotene and β-carotene is correct name.

P7 line 299; with β carotenoids such as cryptoxanthin and astaxanthin,

→with carotenoids such as cryptoxanthin and astaxanthin

Please check.

P10 Table 1 Supplements; β carotenoids

→Related to above, β-carotene is correct. Please check.

Author Response

Reviewer 3

In this manuscript, author reviewed anti-inflammatory strategies targeting metaflammation. This manuscript is well described about this topic. Please consider below.

We thank the Reviewer for the positive reception of our manuscript and drawing attention to the errors in the nomenclature. Following the Reviewer's suggestion, all of them were corrected in the revised version of the manuscript.

P7 line 299; α and β carotenoids [72].

→They are compound name. So, α-carotene and β-carotene is correct name.

P7 line 299; with β carotenoids such as cryptoxanthin and astaxanthin,

→with carotenoids such as cryptoxanthin and astaxanthin

“Similar findings concern increased consumption of foods reach in α and β-carotenes [90]. Moreover, supplementation with carotenoids such as cryptoxanthin and astaxanthin, that exhibit antioxidant and anti-inflammatory effects, and favorable regulate M1/M2 macrophage polarization in the liver, occurred to be effective in prevention and reversal of lipotoxicity-induced hepatic insulin resistance and steatohepatitis in mice. However, there is no evidence that carotenoids exhibit beneficial effects in patients with NAFLD [reviewed in 53].” (line 389-394)

P10 Table 1 Supplements; β carotenoids

→Related to above, β-carotene is correct. Please check.

The suggested correction was introduced in Table 1.

Round 2

Reviewer 2 Report

Significantly improved.